# Einsum Benchmark: Enabling the Development of Next-Generation Tensor Execution Engines

**Mark Blacher**[1][*], **Christoph Staudt**[1], **Julien Klaus**[1], **Maurice Wenig**[1], **Niklas Merk**[1],
**Alexander Breuer**[1], **Max Engel**[1], **Sören Laue**[2], **Joachim Giesen**[1]
[1]University of Jena, [2]University of Hamburg

## Abstract

Modern artificial intelligence and machine learning workflows rely on efficient tensor libraries. However, tuning tensor libraries without considering the actual problems they are meant to execute can lead to a mismatch between expected performance and the actual performance. Einsum libraries are tuned to efficiently execute tensor expressions with only a few, relatively large, dense, floating-point tensors. But, practical applications of einsum cover a much broader range of tensor expressions than those that can currently be executed efficiently. For this reason, we have created a benchmark dataset that encompasses this broad range of tensor expressions, allowing future implementations of einsum to build upon and be evaluated against. In addition, we also provide generators for einsum expressions and converters to einsum expressions in our repository, so that additional data can be generated as needed. The benchmark dataset, the generators and converters are released openly and are publicly available at `https://benchmark.einsum.org`.

## 1  Introduction

Einsum is a powerful interface for describing tensor expressions in a clear and concise way. Although the underlying notation has existed for over a century, einsum itself only gained mainstream recognition in 2011 with its implementation in NumPy [24]. Hence, the idea of using einsum as a computational backend for tensor expressions is relatively new, and mapping computations to einsum is a field with many applications yet to be found. As it turns out, einsum naturally allows serving inference queries on Boolean, semantic, and probabilistic AI models [9]. Moreover, computations involving tensor networks [14, 19, 27, 48, 74] can be easily ported to einsum, because a single einsum expression can describe the contraction of an entire tensor network. The expressive power of einsum led to its growing popularity. Einsum is now part of the most important frameworks for machine learning and artificial intelligence [1, 12, 52] as well as numerous array computing libraries [38, 47]. Einsum is heavily used in deep learning [62, 64], and its notation can serve to describe or even enable new AI models [31, 53, 54]. Einsum backends are also a popular choice for simulating quantum circuits [15], because an arbitrary quantum circuit can be mapped directly to a tensor network contraction problem [36] and thus to einsum.

However, describing a tensor computation succinctly with einsum notation does not mean that it is executed efficiently on the backend that performs the actual computation. For einsum expressions involving more than two tensors, an evaluation order, that is the contraction path, must first be computed so that the expression can be executed with low computational cost. Yet, even an efficient contraction path does not ensure fast execution. The actual performance of an einsum expression heavily relies also on efficiently executing the individual tensor operations using the full potential of the available hardware. This dichotomy, first, finding an efficient contraction path, and second,

---

[*]Corresponding author: mark.blacher@uni-jena.de

38th Conference on Neural Information Processing Systems (NeurIPS 2024) Track on Datasets and Benchmarks.

performing the actual computation, is what makes implementing efficient einsum engines challenging. According to our experimental measurements on the benchmark data, both parts, finding a contraction path and executing the individual tensor operations, need to be tuned to support future einsum workloads.

When NumPy introduced einsum, it was designed for operations involving just one or two tensors, using a single-threaded, nested loop approach without contraction paths. However, the current common practice in Python programming environments is to use `cotengra` [20, 22] for finding contraction paths for challenging problems and `opt_einsum` [66, 68] coupled with a backend such as PyTorch [52] for executing individual tensor operations. Both `cotengra` and `opt_einsum` provide functions to generate random einsum problems, but these randomly generated problems only reflect a small range of practical einsum applications. The number of tensors in these generated problems tends to be rather small if the problems are meant to be executed. Many structures that are common in real-world einsum expressions, such as tensor networks with hyperedges, Hadamard products, tensor diagonals, or traces, are missing in the generated expressions. The same limitations are even more apparent in existing benchmark datasets [39, 70], which predominantly feature structurally simple expressions composed of only two input tensors. However, there are also generators for specific problem classes, such as quantum circuits [16, 35] or tree tensor networks [72, 73], that are capable of generating larger and more complex problems. Nonetheless, the results of these generators must first be converted to einsum expressions. In addition to the generators, there is also benchmark data for quantum supremacy circuits [26, 37], graphical models [75, 76], model counting problems and weighted model counting problems [43, 44, 45, 46]. This benchmark data must also first be converted before it can be executed in einsum.

Our contributions are as follows: First, we introduce a comprehensive einsum benchmark dataset that encapsulates a broad spectrum of real-world and generated problems. Second, we provide generators and converters to create new problem instances. And third, we use instances from our benchmark dataset to demonstrate performance pitfalls of current einsum engines.

## 2 Background

### 2.1 Notation

In 1916, Einstein introduced a notation with which operations involving summations over several tensor indices can be expressed concisely [17]. Hence, the term *einsum*, a blend of *Einstein* and *summation*. However, the notational convention shifted. Originally, Einstein assumed an implicit summation over pairs of identical indices in a tensor expression. In its modern form, the notation explicitly specifies the indices of the output tensor, and the indices that are not part of the output tensor are used for summation [49]. For example, consider the tensor expression $AB^\top v$, where $A$ is a matrix in $\mathbb{R}^{I \times K}$, $B$ is a matrix in $\mathbb{R}^{J \times K}$, and $v$ is a vector in $\mathbb{R}^J$. In Einstein's original notation, the example tensor expression is $A_{ik}B_{jk}v_j$, where the indices $j$ and $k$ appear twice and are thus implicitly used as summation indices. The latter expression is therefore a short form of $\sum_j \sum_k A_{ik}B_{jk}v_j$. In modern einsum notation, the example tensor expression is $A_{ik}B_{jk}v_j \to {}_i$, where $i$ is the index of the output tensor. Modern einsum notation is more powerful than the classical one, because in modern einsum notation, expressions like $A_{ik}B_{jk}v_j \to$ or $A_{ik}B_{jk}v_j \to {}_{ijk}$ are also possible. The first expression evaluates to a scalar, the second to a third order tensor.

Common einsum APIs support both the modern way of specifying tensor expressions using an arrow and the classical one, where summation over repeated indices is implicit. In this work, we stick to the modern way with the arrow. When tensor names are not crucial, the key information of a tensor expression can be succinctly captured using only the indices of the tensors in a simple format string. We use a format string similar to that of the `einsum` function in the NumPy library [8]. The format string representing the example expression $A_{ik}B_{jk}v_j \to {}_i$ is `"ik,jk,j->i"`. Table 1 provides further examples of format strings that illustrate the expressive power of modern einsum notation.

### 2.2 Contraction path

Einsum expressions can contain dozens, thousands or even hundreds of thousands of tensors. It is therefore essential to find an efficient evaluation order for a given einsum expression. Usually, an einsum expression is evaluated in pairs of tensors, that is, only two tensors are contracted, that

Table 1: Examples of einsum format strings for tensor expressions.

| Operation | Format string |
|---|---|
| Matrix trace | `ii->` |
| Matrix diagonal | `ii->i` |
| Vector outer product | `i,j->ij` |
| Mahalanobis distance | `i,ij,j->` |
| Triplestore query [9] | `ij,i,jk->k` |
| Marginalization (sum over multiple axes) | `ijklmnop->m` |
| Batch matrix multiplication | `bik,bkj->bij` |
| Bilinear transformation [40] | `ik,klj,il->ij` |
| Hadamard product (element-wise product) | `ijkl,ijkl->ijkl` |
| Matrix chain multiplication | `ik,kl,lm,mn,nj->ij` |
| $2 \times 3$-tensor network [29] | `ij,iml,lo,jk,kmn,no->` |
| Tucker decomposition [61] | `ijkl,ai,bj,ck,dl->abcd` |
| Weighted model counting [44] | `b,c,d,e,f,ef,eg,bc,cdc->` |
| Graphical model query [60] | `a,ab,bcd,defg,fhi,hj,ik,l->` |
| Tensor regression network [31] | `abcde,fghij,bf,cg,dh,ei,kj->ak` |
| Tensor network language model query [54] | `ab,acd,bef,degh,lk,lci,kjf,ijgh->fh` |
| Max-Cut quantum circuit [16] | `a,b,c,da,eb,fc,ghde,ijgf,klhj,i,k,l->` |

is merged, at a time [55]. Finding an optimal contraction path—a sequence of pairwise tensor contractions that executes an einsum expression at the lowest possible computational cost—is an NP-hard problem [34]. In our example expression $A_{ik}B_{jk}v_j \to_i$, there are two potential sequences for contracting over common indices. Contracting over $k$ first followed by $j$ entails a matrix-matrix multiplication between $A$ and $B$, after which the resulting matrix is multiplied by the vector $v$. Alternatively, contracting $j$ first and then $k$ involves a matrix-vector multiplication between $B$ and $v$, followed by the multiplication of matrix $A$ with the resulting vector. The contraction sequence $j, k$ is the more favorable one here, as it avoids the costly matrix-matrix multiplication.

Representing a contraction path as a sequence of indices is feasible when each index in the path is part of exactly two tensors [42]. However, an index in an einsum expression may be part of more than two tensors, or a pairwise operation may involve outer products where tensors do not share indices. Consequently, einsum libraries adopt a position-based format for the contraction path. In our example, contracting over index $j$ involves contracting the second tensor with the third tensor. Subsequently, contracting over $k$ means contracting the first tensor with the intermediate tensor, which occupies the fourth position, because each intermediate tensor is assigned to the next free position. Assuming we start numbering positions from zero, the contraction path $j, k$ can be represented as a list of position tuples: $[(1, 2), (0, 3)]$. Note that each initial tensor and each intermediate tensor has a unique position. Therefore, this position-based format is referred to as the *static single assignment* (SSA) format. In SSA format, each tensor occupies a static, single position that does not change throughout the computation. An alternative position-based format is the linear format, where the positions of tensors change during computation [66, 67]. The linear format, however, is less efficient.

## 3 The benchmark

In this section, we outline the composition of the benchmark dataset and its associated metadata. We provide detailed information about the actual problems that the respective einsum expressions solve and discuss what makes these einsum expressions particularly interesting for inclusion in a benchmark dataset.

### 3.1 Problem categories

The benchmark dataset consists of 168 einsum problems divided into seven categories. Table 2 shows the categories, the number of problems within each category, the range of tensor counts for problems in each category, and additional details on whether the einsum problems in a given category may include hyperedges, Hadamard products, or repeating indices within a single tensor. Hyperedges means that an einsum problem contains contraction indices that are shared by more than two tensors.

Hadamard products are element-wise multiplication operations performed between two tensors of identical dimensions. And, repeating indices within a single tensor represent either tensor traces or tensor diagonals, as indicated by the indices of the output tensor. Due to their presence, modern einsum libraries should support hyperedges, Hadamard products and traces or diagonals in tensor expressions, both in searching for contraction paths and in executing the actual expressions. However, contraction path algorithms [42, 73] and tensor libraries [56, 70] frequently overlook these features for the sake of simplicity.

Table 2: Benchmark categories with respective problem counts, tensor count ranges, and presence of hyperedges, Hadamard products, and repeating indices within a single tensor (referred to as traces).

| Category | Problems | Tensors | Hyperedges | Hadamards | Traces |
|----------|----------|---------|------------|-----------|--------|
| Graphical models | 10 | 125–3 692 | ✓ | ✗ | ✗ |
| Tensor network language models | 25 | 38–178 | ✗ | ✗ | ✗ |
| Model counting | 50 | 331–579 972 | ✓ | ✓ | ✓ |
| Quantum computing | 32 | 202–17 136 | ✓ | ✓ | ✗ |
| Random problems | 16 | 53–1 668 | ✓ | ✓ | ✓ |
| Structural problems | 21 | 26–2 000 | ✗ | ✗ | ✗ |
| Weighted model counting | 14 | 358–230 848 | ✓ | ✓ | ✓ |

In the following, we present the individual categories according to which we divide the einsum problems.

**Graphical models:** Graphical models are sparse representations of multivariate probability distributions. Graphical models are highly valued for their flexibility in answering inference queries, such as computing probabilities like $P(X = x | E = e)$, where $X$ represents a set of query variables and $E$ denotes a set of evidence variables. Such inference queries are used for prediction, classification, and decision-making across diverse fields such as bioinformatics, social science, and artificial intelligence [30, 41, 79]. Graphical models with discrete variables are dual to tensor networks [60]. Therefore einsum is a viable backend for answering inference queries on these type of models. For the einsum benchmark, we convert ten challenging computations from the *UAI 2022 Competition*, which focuses on inference tasks on graphical models [75, 76].

**Tensor network language models:** Systems with long distance correlations such as natural languages can be modeled via quantum pure states, which describe a probability distribution over a fixed-size context of tokens [54]. These quantum pure states can be represented by a tensor network, where the structure of the tensor network determines the pairwise correlations of different positions in the context. What sets this architecture apart from transformer-based models is that inference queries can be fully expressed using einsum. For the benchmark, we include seven inference queries, which compute the joint probability distribution of two tokens in the context, and 18 queries used in the learning process, which compute the likelihoods for a batch of given contexts.

**Model counting:** Counting the number of solutions, or satisfying assignments of truth values, to a given Boolean satisfiability problem (SAT) in conjunctive normal form is called model counting. Model counting is also known as the #SAT problem, which is #P-complete [77, 78]. Note that, solving a #SAT problem not only determines the solvability of the corresponding SAT problem but also facilitates solving the MAJ-SAT problem, which asks whether the majority of assignments satisfy the formula [3, 28]. For the benchmark, we use 36 model counting problems from four model counting competitions [43, 44, 45, 46], 15 of which we simplify using Arjun [6, 69]. Additionally, we include 14 model counting problems from a benchmark originally designed to sample satisfying assignments of given SAT formulas [23].

**Quantum computing:** By leveraging quantum mechanics principles such as entanglement and superposition, quantum computing aims to eventually achieve computational speeds that surpass those of classical computers [7]. However, as quantum computers are still in their infancy, classical computers are used to simulate quantum computations. Einsum is a promising backend for efficiently simulating quantum computations [15, 36]. The benchmark dataset contains 32 quantum circuits as einsum expressions, ranging from quantum supremacy [26, 37] and Max-Cut circuits [16] to quantum Fourier transform and variational circuits [35].

**Random problems:** For generating random einsum problems, we use the function `rand_equation` from `opt_einsum` [66] and our own generator. With `opt_einsum` we generate four problems, with

our generator, eleven problems. The einsum problems created by our generator are particularly challenging because they are completely unstructured, have different dimension sizes and always contain hyperedges and tensor traces or tensor diagonals.

**Structural problems:** Structural problems are defined by their fixed, predefined tensor network architectures [51]. For structural problems belonging to the class of tree tensor networks, an optimal contraction path can be computed in $\mathcal{O}(n^3)$ time, where $n$ is the number of input tensors in an einsum expression [73]. Tree tensor networks are well suited for approximating multivariate functions [4, 5], or simulating high-dimensional quantum systems [18]. Tree tensor networks are represented in our benchmark by Fork Tensor-Product State (FTPS) problems and matrix chain multiplication problems. Additionally, the benchmark includes Multi-Scale Entanglement Renormalization Ansatz (MERA) problems, Projected Entangled-Pair State (PEPS) problems [72], and problems involving inner products between two Matrix Product States (MPS) [66]. Moreover, we use Cotengra's generators [20] to create regular graph problems and lattice problems. The benchmark data set contains a total of 21 structural problems.

**Weighted model counting:** Weighted model counting extends classical model counting by weighting the literals in propositional (SAT) formulas, allowing it to sum the weights of all satisfying assignments. This capability enables handling complex probabilistic scenarios where outcomes have varying likelihoods, making weighted model counting ideal for tasks involving probabilistic reasoning, risk analysis, and decision-making under uncertainty [13, 63]. Also, a weighted model counting engine, and thus einsum, can serve as a backend for probabilistic programming languages [25]. For the benchmark we use 14 problems from four weighted model counting competitions [43, 44, 45, 46].

### 3.2 Metadata

The benchmark dataset includes an accompanying CSV table that provides additional metadata for each problem in the dataset. Based on the metadata, problems can be selected according to the desired characteristics. The metadata are also helpful in understanding which types of problems exist. In the benchmark repository, we also include code to generate metadata for new einsum problems, allowing users to explore the characteristics of einsum expressions that are not present in the dataset.

We present the provided metadata by using the weighted model counting einsum expression `"b,c,d,e,f,ef,eg,bc,cdc->"` from Table 1. The metadata attributes and their values for this expression are shown in Table 3, with each attribute's value supplemented by an explanation. Note that, although the expression contains six unique indices (b, c, d, e, f, g), the index g is not a contraction index because it only appears once in the expression. Efficient einsum libraries eliminate indices that appear only once in an expression or indices with a dimension size of one before beginning to contract the expression [21, 55]. The presence of such an optimization potential in an einsum expression can be derived from its metadata attributes: different indices, contraction edges, and smallest dimension size. Note also, that an einsum expression can be split into independent components if the components have no common contraction indices. These components can be handled separately for finding efficient contraction paths, or for executing the expression.

Table 3: Metadata for `"b,c,d,e,f,ef,eg,bc,cdc->"`. All indices are of dimension size two.

| Metadata attributes | Value | Explanation |
|---|---|---|
| Input tensors | 9 | The expression contains nine input tensors. |
| Different indices | 6 | Six unique indices are utilized across all tensors. |
| Hadamard products | 0 | There are no Hadamard products in the expression. |
| Contraction edges | 5 | Five indices (b, c, d, e, f) are contraction indices. |
| Contraction hyperedges | 2 | Two contraction indices (c, e) are hyperedges. |
| Tensors in largest hyperedge | 3 | The largest hyperedge contains three tensors (c, bc, cd). |
| Tensors with traces or diagonals | 1 | One tensor (cdc) contains repeating indices. |
| Independent components | 2 | There are two components (e, f, ef, eg and b, c, d, bc, cdc). |
| Tensors in largest component | 5 | The largest component has five tensors (b, c, d, bc, cdc). |
| Smallest dimension size | 2 | The smallest dimension size across all indices is two. |
| Largest dimension size | 2 | The largest dimension size across all indices is two. |
| $\log_2(\text{output size})$ | 0 | The output is a scalar, hence $\log_2(1) = 0$. |

In addition to the twelve metadata attributes and their corresponding values, we provide two contraction paths for each einsum problem in the benchmark: one optimized to minimize the number of operations, and the other to minimize the size of the largest intermediate tensor. It is generally assumed that minimizing the number of operations leads to faster execution of the actual expression [42, 55], whereas minimizing the intermediate size reduces the amount of RAM required to execute the expression. However, this is an oversimplification, particularly concerning the execution time required for an einsum expression (see Section 4.2). Nevertheless, optimizing contraction paths for the number of operations or intermediate size remains standard practice, as better alternative optimization metrics are currently not available. Therefore, we compute two example contraction paths for each problem in the benchmark: one aimed at minimizing the number of operations, and another targeted at reducing the size of the largest intermediate tensor. For computing these paths, we use our graph partitioning strategy [71] in conjunction with a greedy approach that incorporates multiple cost functions [10, 50]. The code for computing contraction paths is included in our repository, enabling users to compute paths for new einsum problems.

Each benchmark problem is stored in one pickle file [57], containing the einsum format string, the NumPy input tensors, the sum of all values in the output tensor, and the two contraction paths. Additionally, each contraction path is accompanied by its metadata: the base-10 logarithm of the required number of operations, the base-2 logarithm of the largest intermediate tensor size, the lowest intermediate density of a tensor, and the average density across all intermediate tensors. The sum of all values in the output tensor helps to verify the accuracy of the computed result. The lowest intermediate density reveals the sparsest tensor produced in a pairwise contraction, whereas a low average density across all intermediate tensors suggests that a sparse einsum implementation might be better suited to efficiently solve the problem.

# 4 Experiments

With a dataset comprising various einsum problems, we can evaluate the performance of current einsum libraries. In total, we conduct five experiments to uncover performance pitfalls of current einsum implementations. The first four experiments are carried out exclusively on the CPU, while the fifth experiment additionally includes measurements on the GPU. All subsequent experiments are conducted on a machine featuring an Intel i9-10980XE 18-core processor, running Ubuntu 20.04.6 LTS, with 128 GB of RAM. Each core operates at a base frequency of 3.0 GHz, reaches up to 4.6 GHz in turbo mode, and supports the AVX-512 vector instruction set. Additionally, the system is equipped with a Quadro RTX 4000 GPU with 8 GB of GDDR6 SDRAM. For the experiments, we use Python 3.10.9 with the following packages: `opt_einsum` 3.3.0, numpy 1.26.4, `torch` 1.12.1, `tensorflow` 2.16.1, `jax` 0.4.28, `jaxlib` 0.4.28, `cotengra` 0.6.2, `kahypar` 1.3.5, and `sqlite` 3.45.3.

## 4.1 Scalability of processing large contraction paths

Before executing an einsum expression, a contraction path must be computed. The question we address here is whether this path computation is scalable to large problem sizes. In the following experiment, we compute a single contraction path using `opt_einsum` and its greedy algorithm. We chose `opt_einsum` over `cotengra` because we found that `opt_einsum` is significantly faster at computing a single path. We perform the path computations for five einsum expressions of varying sizes, ranging from 653 to 579 972 tensors. Additionally, we use PyInstrument [59] to profile each path computation, enabling us to identify the compute-intensive parts of the computation. Figure 1 shows the results of the experiment.

As einsum expressions increase in scale, surprising inefficiencies in `opt_einsum` become apparent. After a contraction path is computed, generating the strings that describe the pairwise computations emerges as a severe bottleneck as the expressions grow larger. Additionally, the conversion from the internally used SSA path format to the less efficient linear path format, which serves as the output format, starts to overshadow the time actually spent on computing a valid path. Such avoidable inefficiencies result in unnecessarily long path computation times for larger einsum expressions.

Similar inefficiencies occur even when a precomputed contraction path is passed to `opt_einsum` for executing an expression. Internally, `opt_einsum`, here also inefficiently generates the strings that represent pairwise contractions, resulting in overhead that can dominate the execution time for large einsum expressions. Furthermore, `opt_einsum` executes the expressions according to the

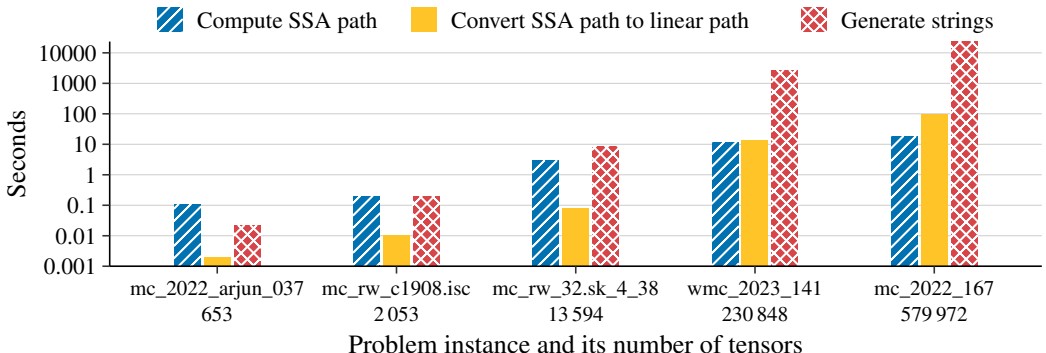

Figure 1: Computations of a single contraction path for five instances of the dataset, sorted by the number of tensors in the expressions. For each instance, the time for computing a single path is divided into its three constituents: computing the actual path, and overhead operations such as path conversion to the linear format and string generation. Times are shown on a logarithmic scale.

provided linear path, which involves steadily removing tensors from arbitrary positions in the list of all tensors and appending new intermediate tensors to the list. This method of execution is particularly inefficient for large einsum expressions. Therefore, due to inherent inefficiencies, einsum libraries should completely avoid the linear path format and adopt the SSA format across all library levels.

## 4.2  Asymmetric tensor sizes in pairwise contractions

When executing an einsum expression along a specified contraction path, current libraries map the individual pairwise tensor contractions to highly efficient linear algebra or tensor library calls. For example, PyTorch maps pairwise tensor contractions to its tensor library, ATen [58], whereas `opt_einsum` maps individual pairwise tensor contractions to standard BLAS calls [11] of the respective backend. However, such tensor contractions only achieve peak performance when both involved tensors are large and their shapes are not skewed [70]. For example, performing a matrix-matrix multiplication $AB$, where matrix $A$ is in $\mathbb{R}^{2\times2}$ and matrix $B$ in $\mathbb{R}^{2\times2^{30}}$, is highly inefficient with current libraries. However, these types of contractions are abundant, particularly in quantum computing problems. Ironically, optimizing the contraction path to minimize the number of operations often leads to skewed tensor contractions during the execution of the expression [26]. To demonstrate that the number of theoretical operations serves only as a relatively weak estimator of the execution time for an einsum expression, we compute 200 contraction paths using `cotengra` for the quantum computing problem `qc_circuit_n49_m14_s9_e6_pEFGH_simplified`. Among these contraction paths, 100 paths are optimized to minimize the number of operations, whereas the remaining 100 paths aim to minimize the size of the intermediate tensor. Figure 2 shows the execution times for the quantum computing problem across all computed paths using `opt_einsum` with a PyTorch backend.

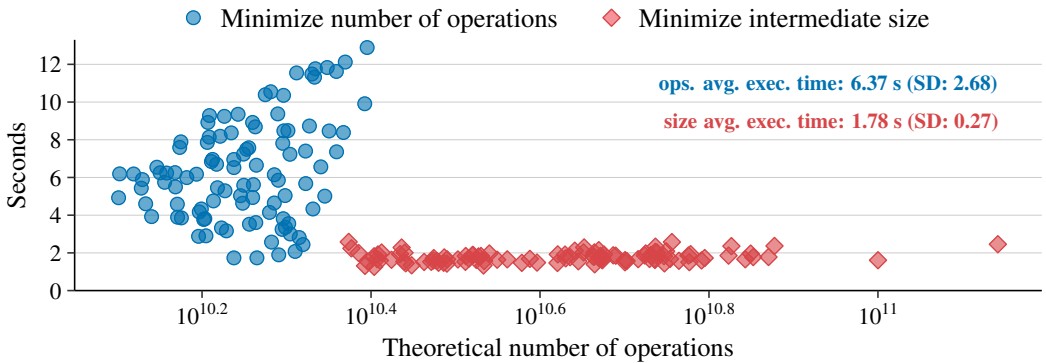

Figure 2: Execution times in seconds for `qc_circuit_n49_m14_s9_e6_pEFGH_simplified` across 200 contraction paths, sorted by the number of operations required for running the expression.

Minimizing the intermediate tensor size when computing contraction paths can lead to more balanced tensor contractions and, consequently, lower average execution times compared to paths optimized to minimize the theoretical number of operations. Therefore, focusing solely on reducing the number of operations, without accounting for the skewness of resulting pairwise contractions, can compromise performance. Also, neglecting to optimize linear algebra and tensor libraries for skewed contractions is a significant oversight, given their frequent occurrence. It is important to note that optimizing skewed tensor contractions relates to minimizing data movements across the memory hierarchy and between processors [32, 33]. Einsum expressions involving many skewed tensor contractions can lead to increased data movements, thereby becoming memory-bound and exhibiting poor cache utilization. Both aspects—accounting for skewness in computing contraction paths and enhancing the speed of skewed tensor contractions—must be addressed to reduce data movements and thus improve execution times of einsum expressions.

## 4.3 Sparse intermediate tensors

Input tensors containing one or more zeros can result in highly sparse intermediate tensors. Initially, these problems may appear dense, but they can become sparse while evaluating the expression. In the benchmark, this phenomenon is particularly evident for model counting and weighted model counting problems, as well as for certain quantum computing problems. For the sparse experiment, we chose the model counting einsum problem `mc_2021_036`, characterized by an average density of $5.45 \times 10^{-4}$, indicating that 0.0545 % of the entries in all intermediate tensors are non-zero. Executing this problem via `opt_einsum` using the PyTorch backend turns out to be slower compared to performing the entire computation in SQLite (see Table 4).

Table 4: Average execution times in seconds and standard deviations for ten runs of the sparse model counting einsum problem `mc_2021_036` using PyTorch and SQLite.

| Backend | Tensor format | Execution | Time (seconds) |
|---------|---------------|-----------|----------------|
| PyTorch | Dense | Multi-threaded | $43.2 \pm 0.09$ |
| SQLite | Sparse | Single-threaded | $34.2 \pm 0.04$ |

While PyTorch's einsum supports only dense tensors, the SQL implementation employs a sparse tensor storage format, specifically the coordinate format [9]. However, mapping einsum problems to SQL is rather inefficient, especially when the dimensions of the problem are small, because for each non-zero value, all coordinates of its position must be stored explicitly. In the example model counting problem, all dimensions are of size two. Also, note that SQLite executes the einsum query single-threaded. Nonetheless, the fact that SQLite outperforms PyTorch in executing the example problem clearly signals that incorporating sparsity-aware algorithms into einsum libraries could significantly enhance their efficiency.

## 4.4 Data types

The tensors used in the einsum expressions of the benchmark are of various data types, including `int16`, `int32`, `int64`, `float32`, `float64`, `complex64`, and `complex128`. However, all tensors within a particular expression are always of the same data type. The AVX-512 CPU used for our measurements is equipped with instructions for vectorized addition and multiplication for `int16`, `int32`, `int64`, `float32`, and `float64`. Additionally, for `float32` and `float64`, there are fused multiply-add (FMA) instructions that combine multiplication and addition into a single instruction, enhancing computational efficiency for floating-point operations. An efficient einsum library is expected to exhibit performance variations between data types approximately proportional to the size differences of the data types. For example, tensor computations using an `int16` data type are expected to be four times as fast as those using an `int64` data type. This expectation rests on the assumption that smaller data types consume less memory bandwidth and accommodate more values within vector registers compared to larger data types. To investigate whether this expectation holds true in practice, we select one `int16` einsum problem (`mc_rw_blockmap_05_01.net`) and convert it to all data types that are sufficiently large to retain the exact solution. Table 5 shows the execution times for this expression across the various data types and `opt_einsum` backends.

Table 5: Average execution times in seconds and standard deviations for ten runs of the einsum problem `mc_rw_blockmap_05_01.net` across various data types and backends. This problem involves 2 737 input tensors, has a maximum intermediate tensor size of $2^{28}$, and requires $10^{11.33}$ operations. Entries labeled as N/A indicate that the backend is unable to compute a valid result.

| Backend | int16 | int32 | int64 | float32 | float64 | complex64 | complex128 |
|---|---|---|---|---|---|---|---|
| NumPy[a] | N/A | N/A | N/A | N/A | N/A | N/A | N/A |
| PyTorch[b] | N/A | N/A | $41.77 \pm 0.38$ | $5.18 \pm 0.18$ | $7.93 \pm 0.29$ | $8.16 \pm 0.29$ | $13.78 \pm 0.24$ |
| TensorFlow[c] | N/A | $41.58 \pm 0.17$ | $44.45 \pm 0.13$ | N/A | $42.30 \pm 0.11$ | $43.59 \pm 0.15$ | $47.89 \pm 0.09$ |
| JAX[d] | $220.18 \pm 0.91$ | $25.27 \pm 0.13$ | N/A | $19.36 \pm 0.08$ | N/A | $31.91 \pm 0.07$ | N/A |

[a]Error: *too many subscripts in einsum*, caused in NumPy by exceeding the maximum of 32 different indices.
[b]Scalar type errors due to hidden conversions from `int16` and `int32` to `int64` during pairwise contractions.
[c]No `int16` support. For `float32`, it only supports tensors with a rank $\leq 12$, but this problem requires rank 28.
[d]Truncates `int64` to `int32`, `float64` to `float32`, and `complex128` to `complex64`.

None of the `opt_einsum` backends can execute the example expression for all data types natively, despite hardware support. For floating-point data types, including complex types, PyTorch consistently demonstrates superior performance and exhibits robust performance scaling. Conversely, JAX struggles with the `int16` data type, while NumPy is unable to process any data type due to its index limitation. TensorFlow, on the other hand, shows some degree of performance scaling across the different data types. Nonetheless, the substantial performance gap between TensorFlow and PyTorch indicates significant inefficiencies in TensorFlow's execution of pairwise einsum expressions.

## 4.5 Hybrid CPU-GPU execution

GPUs are known for their superior performance in executing large tensor contractions compared to CPUs [2, 65]. However, when executing many small tensor contractions, GPUs may be slower than CPUs. In these cases, the overhead of launching GPU processes and managing resources can outweigh the benefits for smaller tasks, and the operations may not fully utilize the GPU's parallel processing capabilities. Additionally, frequent data transfers and the complexity of scheduling and synchronizing these tasks can significantly reduce the performance advantages of GPUs, making CPUs a more efficient option for handling small tensor contractions.

Current einsum implementations require deciding upfront the execution policy between CPU and GPU for a tensor expression. Hybrid CPU-GPU execution, such as running small contractions on the CPU and large ones on the GPU, is not supported. In this experiment, we demonstrate that a hybrid CPU-GPU execution for tensor contractions may outperform CPU-only and GPU-only execution options. We chose the `float64` model counting problem `mc_2020_082`, which involves $10^{10.67}$ theoretical operations and comprises 195 372 tensors. We set a threshold of 1024 entries in a tensor for switching from CPU to GPU. This means that as soon as at least one tensor in the pairwise contraction exceeds 1024 entries, the operation is performed on the GPU. If not, the operation is executed on the CPU. Once a tensor is on the GPU, all subsequent contractions involving it also occur on the GPU to avoid transferring the data back to the CPU. As the computational backend, we use PyTorch. We execute the pairwise tensor contractions using a simple for-loop over the SSA path because relying on `opt_einsum` alone would not allow us to test the proposed hybrid execution policy. Additionally, using `opt_einsum` for large tensor expressions would be highly inefficient due to its string generation and path conversion bottlenecks, as demonstrated in the first experiment (see Section 4.1). Table 6 shows the execution times for the three execution policies: CPU, GPU, and CPU-GPU.

Table 6: Average execution times in seconds and standard deviations for ten runs of the model counting problem `mc_2020_082` across different execution policies. The GPU (%) column shows the percentage of tensor contractions performed on the GPU for each policy.

| Execution policy | GPU (%) | Time (seconds) | Speedup |
|---|---|---|---|
| CPU | 0.00 | $4.71 \pm 0.03$ | Baseline |
| GPU | 100.00 | $6.48 \pm 0.04$ | 0.73x |
| CPU-GPU | 2.62 | $3.28 \pm 0.01$ | 1.44x |

Using the GPU alone for the example model counting problem results in the lowest performance. In contrast, the hybrid CPU-GPU execution policy proves to be the fastest option, indicating that `einsum` could benefit from fine-grained hybrid execution strategies.

## 5 Limitations

We only included problems in the dataset that start small and become large in the course of evaluating the einsum expression. Otherwise, the benchmark dataset would quickly exceed the limits of common storage devices and require long read times for the data, which would reduce its practicability. We also avoided including problems that are too big to compute on a single compute node, because before even considering tuning distributed einsum implementations, shared memory einsum implementations must be optimized first. Otherwise, the performance pitfalls of shared memory einsum implementations propagate to distributed settings. However, as needed, such storage-hungry, compute-intensive or RAM-hungry instances can still be created using the tools provided in our repository.

In our benchmark, we excluded tensor expressions with trivial contraction paths, specifically those involving only a few tensors. Such expressions are abundant in current einsum workloads, and state-of-the-art tensor libraries demonstrate satisfying performance on these expressions in achieving almost theoretical peak performance in terms of floating-point operations per second for compute-bound tensor contractions [39, 70].

Our benchmark dataset is also limited in terms of the data types for which we provide einsum problems. The benchmark primarily focuses on portable primitive data types to ensure compatibility across a wide range of devices. However, if users require specialized data types, they can convert them from the available types, provided the conversion maintains sufficient accuracy. Additionally, the generators in the repository can be used to create new einsum expressions, which can then be initialized with tensors that produce valid results for specialized data types. For example, using this approach, TensorFloat-32 einsum problems can be generated for execution on an NVIDIA GPU.

## 6 Conclusions

This paper introduces an einsum benchmark dataset that addresses the lack of publicly available problems for evaluating einsum libraries. Einsum is pivotal as a computational backend in AI/ML and physics applications. Frontend problem descriptions are usually converted into backend einsum expressions at runtime and are lost as soon as the computation finishes. Therefore, challenging real-world einsum expressions are rarely available for developers who are tuning their libraries. We gathered and combined a diverse set of expressions into a single benchmark and enriched it with additional expressions from publicly available generators or our own generators, so that library developers have a wide range of einsum problems at their disposal. In the experiments it became clear that such instances are needed to improve current einsum libraries. In particular, einsum libraries face scalability issues when computing or processing long contraction paths, perform poorly with highly asymmetric tensor sizes, ignore sparse intermediate tensors, struggle with integer data types, and miss the opportunity for hybrid CPU-GPU execution.

Looking ahead, it is foreseeable that einsum expressions will grow increasingly complex and lengthy, and new ML models leveraging einsum will emerge. Thus, it is crucial to start adapting einsum libraries now to ensure that they perform well on current and future workloads. This benchmark dataset and the tools we provide in our repository are designed to facilitate this adaptation, providing a resource to assist developers in enhancing library capabilities to meet current and forthcoming computational challenges. However, we believe that there are other interesting use cases of einsum in other domains that are missing in our benchmark dataset. We encourage anyone who thinks that they have interesting einsum problems to contact us so that we can extend the benchmark dataset.

## Acknowledgments

This work was supported by the Carl Zeiss Foundation within the project *Interactive Inference*.

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
