# Supplementary Material: Einsum Benchmark

**Mark Blacher**[1]*, **Christoph Staudt**[1], **Julien Klaus**[1], **Maurice Wenig**[1], **Niklas Merk**[1],
**Alexander Breuer**[1], **Max Engel**[1], **Sören Laue**[2], **Joachim Giesen**[1]
[1]University of Jena, [2]University of Hamburg

## 1 Motivation

**For what purpose was the dataset created?** The dataset was created with two primary purposes. First, it serves as a benchmark for einsum libraries, enabling the assessment of both the efficiency in determining contraction paths and the performance in executing einsum expressions. Second, it provides developers with a diverse set of einsum problem instances, thereby facilitating the development of more efficient, general-purpose einsum libraries.

**Who created the dataset?** The dataset instances were created by the authors. See also Section 3 on the data collection process.

**Who funded the creation of the dataset?** This work was supported by the Carl Zeiss Stiftung within the project *Interactive Inference*.

## 2 Composition

**What do the instances that comprise the dataset represent?** The instances in the dataset represent tensor expressions from various application domains that can be evaluated via einsum.

**How many instances are there in total?** The benchmark dataset consists of 168 einsum problems, divided into seven categories: graphical models (10), tensor network language models (25), model counting (50), quantum computing (32), random problems (16), structural problems (21), and weighted model counting (14).

**Does the dataset contain all possible instances or is it a sample of instances from a larger set?** There is an infinite number of tensor expressions that can be computed via einsum. The benchmark dataset represents a subsample of this vast expression space.

**What data does each instance consist of?** Each instance consists of the einsum format string that represents the tensor expression, along with the input tensors, and the sum of all values in the output tensor. Additionally, each instance includes two contraction paths: one optimized to minimize the number of operations, and another aimed at reducing the size of the largest intermediate tensor. Furthermore, each contraction path is accompanied by its metadata: the base-10 logarithm of the required number of operations, the base-2 logarithm of the largest intermediate tensor size, the lowest intermediate tensor density, and the average density across all intermediate tensors.

**Is the dataset self-contained?** The dataset is self-contained and does not rely on external resources.

**Does the dataset contain data that might be considered confidential?** The dataset does not contain any confidential data.

---

*Corresponding author: mark.blacher@uni-jena.de

38th Conference on Neural Information Processing Systems (NeurIPS 2024) Track on Datasets and Benchmarks.

**Does the dataset contain personally identifiable information or offensive content?** The einsum tensor expressions in the dataset are purely mathematical and contain neither personally identifiable information nor offensive content. This ensures that the dataset does not cause potential negative societal impacts.

# 3 Collection Process

**How was the data associated with each instance acquired?** Tensor network language model instances were created by us as we explored new machine learning architectures. The instances in the categories of random and structural problems were generated either by our own generators or through external generators, whereas instances from the categories of graphical models, model counting, and weighted model counting were converted from publicly accessible problems provided in other formats into einsum expressions. Quantum computing problems were both generated and converted. All tools used, including generators and converters, are either directly available in our repository or accessible through links provided there.

**What mechanisms or procedures were used to collect the data?** The strategy for generating or converting the data involved creating hundreds or even thousands of instances for each category. Then computing paths and metadata for them, retaining only those instances that were diverse and challenging, yet feasible to execute. For the converted problems, we verified correctness by either using provided solutions or by executing the problems with non-einsum solvers to ensure that the einsum expressions yielded accurate results.

**If the dataset is a sample from a larger set, what was the sampling strategy?** The sampling strategy involved manually inspecting collected data instances and their metadata to select a diverse set of problems that represent the complexity of potential einsum expressions. Consequently, the final selection includes expressions featuring hyperedges, Hadamard products, and repeated indices within a single tensor. The dataset covers expressions ranging from a few dozen to hundreds of thousands of tensors, spanning both floating-point and non-floating-point data types. It includes expressions with independent components and varying dimension sizes. Additionally, before including expressions in the dataset, we executed them to identify any execution peculiarities, such as asymmetric tensor sizes in pairwise contractions or sparsity in intermediate tensors, ensuring that instances with these peculiarities are also represented in the dataset.

**Who was involved in the data collection process?** The data collection of einsum expressions for the benchmark dataset was done by the authors. However, as we also utilized external data generators and converted problems from other formats into einsum expressions, the broader data collection process included those individuals whose data was transformed. These secondary data collectors are duly acknowledged in our dataset repository.

**Over what time frame was the data collected?** The einsum expressions were collected over the time period from September 2023 to May 2024.

# 4 Preprocessing

**Was any preprocessing of the data done?** Einsum expressions in the dataset, representing problems from graphical models, model counting, and weighted model counting, were converted from publicly accessible problems available in other formats [8, 9, 10, 11, 14, 15]. Quantum computing problems [4, 7] were converted using methods provided by quantum simulation software [3, 5, 6]. Furthermore, we simplified 15 model counting problems before converting them to einsum expressions [1, 13].

**Was the raw data saved in addition to the preprocessed?** The raw problems, prior to preprocessing, are provided online in the original data repositories [4, 7, 8, 9, 10, 11, 14, 15]. In naming the instances, we have retained the original names to facilitate identification of the raw problems.

**Is the software that was used to preprocess the data available?**    The converters for graphical models, model counting, and weighted model counting problems are included in our benchmark repository. The software used for converting quantum problems to einsum expressions and for simplifying model counting problems is also freely accessible [1, 3, 5, 6, 13].

## 5    Uses

**Has the dataset been used for any tasks already?**    Parts of the dataset have been utilized in the development of a greedy algorithm aimed at identifying efficient contraction paths [2, 12].

**Is there a repository that links to any or all papers or systems that use the dataset?**    We will include references to projects utilizing the dataset within our repository.

**What other tasks could the dataset be used for?**    In addition to benchmarking einsum libraries and developing more efficient ones, this dataset could be used for comparing and developing mapping strategies to einsum expressions. Typically, multiple mapping strategies exist for aligning computational problems with einsum, and identifying particularly efficient mappings is crucial for speeding up einsum computations. The dataset, alongside referenced literature and the converters in the benchmark repository, could serve as a starting point for exploring and thus improving einsum mappings.

## 6    Distribution

**How will the dataset will be distributed?**    The website for the dataset, `https://benchmark.einsum.org`, provides permanent access to the data and experiments. On this site, we include the links to the actual repositories. Utilizing a permanent website allows us to switch repository providers if necessary in the future. Additionally, the dataset is linked to the following DOI: `https://doi.org/10.5281/zenodo.11477304`.

**When will the dataset be distributed?**    Distribution of the dataset commenced in June 2024.

**Will the dataset be distributed under a copyright or other intellectual property license?**    The dataset is distributed under the CC BY 4.0 license, which allows users to distribute, remix, adapt, and build upon the material in any medium or format, including for commercial purposes, as long as they credit the creator. We, the authors, confirm that the dataset does not infringe on any rights. We bear all responsibility in the case of any rights violations related to the publication of this dataset.

**Have any third parties imposed intellectual property-based or other restrictions on the data associated with the instances?**    The data used to derive instances for our benchmark dataset are openly and publicly available under permissive licenses that are compatible with the CC BY 4.0 license governing our dataset. All sources used in the derivation of benchmark instances are duly cited in the licensing documentation maintained in our repository.

## 7    Maintenance

**Who is maintaining the dataset?**    The dataset is being maintained by the *Theoretical Computer Science II* group at the University of Jena.

**How can the manager of the dataset be contacted?**    The manager of the dataset can be contacted via email at mark.blacher@uni-jena.de.

**Is there an erratum?**    Errors will be documented in the update history of the dataset, available on our website. Should any errors be discovered, the dataset will be promptly updated in the repository.

**Will the dataset be updated?** The core dataset we presented will only be updated in case of errors. However, we plan to offer an extended version as we anticipate discovering new and interesting use cases for einsum, or finding new challenging einsum expressions. This extension will be maintained as a separate directory from the core instances.

**Will older versions of the dataset continue to be supported?** Versioning of the dataset will be maintained in the repository. However, the core dataset is intended to remain permanent and unchanged to ensure consistent comparability of research results when executing the einsum instances.

**If others want to contribute to the dataset, is there a mechanism for them to do so?** We welcome contributions to the extended version of the dataset, especially, new and challenging einsum problems that are not included in the core dataset. The best way to contribute is to contact us directly, briefly describe the problem being computed, and provide us with example instances. Contributors will be duly acknowledged in the description of the extended version of the dataset.