# OpenReview forum: "Einsum Benchmark: Enabling the Development of Next-Generation Tensor Execution Engines"
_NeurIPS.cc/2024/Datasets_and_Benchmarks_Track — NeurIPS 2024 Track Datasets and Benchmarks Poster_

### Official Review · Reviewer_3ZT8 · 2024-07-23
**A good benchmark for an important problem**

**Rating:** 7
**Confidence:** 3
**Correctness:** The claims made in the submission app…
**Clarity:** The paper is well written.

**Review:**

This is a solid work studying an important problem: carefully constructing a benchmark that studies the efficiency of various einsum operations on {vendor implementation} x {hardware platform} is of high value and interest to this venue. The commendable parts include the breadth of the problems that the benchmark covers, from language models to quantum computing, and the completeness of the open-source, from the raw data to the code that can generate more data, or replicate the results in the paper. I recommend acceptance, along with a few areas of feedback: the work could benefit much from including execution of much higher interest as part of the benchmark, especially with respect to language models, and results from running einsum operations on GPUs.

**Strengths:**

* Work is potentially high impact and studies an important problem
* Solid presentation, manuscript is easy to follow
* The dataset is extensive and the experimental design are thorough
* Interesting observations regarding 1. effect of contraction paths on performance and 2. asymmetry in tensor sizes

**Additional Feedback:**

No additional feedback.

**Documentation:**

The code and data is available and well-documented.

**Ethics:**

I don't see any ethical concern.

**Limitations:**

The authors have adequately addressed the limitations of their work.

**Opportunities For Improvement:**

1. From inspecting the implementations, it appears that the data for language models are synthetically generated. Considering the impact of language model inference / training performance, it would make more sense to capture real traces from a popular, open architecture. It could be possible that I misunderstood this, and in such case I would appreciate the authors explaining how "seven inference queries" is derived (L 139)
2. The benchmark is only run on a single hardware platform. It would be much more interesting to compare einsum performances on GPUs, or across different platforms. (e.g. turning different vector instruction sets on/off, using different backend implementations for each libraries)

**Relation To Prior Work:**

I am not aware of a comprehensive benchmark on einsum operations. However, the authors should consider providing a brief overview of existing benchmarks on tensor libraries and if any concerns with einsum operations.

**Summary And Contributions:**

The work proposes a benchmark dataset of einsum operations covering a wide range of tensor expressions that appear in real-world problems, supplemented with preliminary benchmarks, data generators and converters that can be used to create more data. The experiments uncover many optimization and compatibility improvement opportunities in popular einsum libraries (e.g. NumPy, PyTorch, TensorFlow, JAX), from scalability of processing large contraction paths, discrepancy in tensor sizes in some of the contraction steps, sparsity of certain tensors during an operation, and datatype support & coercion. The work overall provides insights into how tensor processing libraries should be benchmarked and improved upon, considering their vital role in many AI/ML workloads.

---

> ### Author Rebuttal · Authors · 2024-08-14
>
> Thank you for your insightful feedback. Here is how we plan to address the points you have raised:
>
> **(1) Explaining how "seven inference queries" are derived.**
>
> The seven inference queries are based on the "Tensor Network Language Model" as detailed in Pestun and Vlassopoulos, 2017 [1]. Unfortunately, we initially omitted this reference in our paper. Thank you for pointing out this oversight. We apologize for the confusion and will correct this in our revision.
> We opted for this model over the more popular transformer-based models because the computational complexity of einsum queries in transformers is generally lower, owing to their design of alternating linear and non-linear layers. Conversely, the queries from the tensor network language model can be fully expressed using einsum, showcasing einsum's capability to support innovative ML model architectures.
> These seven inference queries specifically compute the joint probability distribution of the first and last tokens across various model sizes.
> To clarify any misunderstandings, we will revise the description of language models in Section 3.1 to explicitly state that these queries originate from a non-transformer-based model. Additionally, we will consistently use the term "tensor network language model" throughout the paper and in the supplement to avoid any confusion with transformer-based language models.
>
> **(2) GPU experiment. (Note: This suggestion is similar to one raised by Reviewer GRgD)**
>
> Yes, insights into the performance of einsum expressions on GPUs would complete the experiments section. We will add a fifth experiment to the paper that specifically targets performance pitfalls on GPUs. From our work with einsum, we know that although a single matrix-matrix multiplication often performs better on a dedicated GPU than on a CPU, executing the entire einsum expression on a GPU can be significantly slower. We will investigate this peculiarity further in an experiment.
>
> **(3) Prior work in benchmarking einsum. (Note: This suggestion is similar to one raised by Reviewer GRgD)**
>
> Existing benchmarks have primarily focused on structurally simple random einsum expressions (opt_einsum [2]) or those involving only a few random tensors (TBLIS [3], GETT [4]). Additionally, there is notable benchmarking related to contraction path computations, particularly within the realm of quantum computing, which primarily assesses theoretical metrics rather than execution. We have integrated instances from these quantum computing benchmarks or used their generation methods [5, 6]. In the final version of our paper, we will clearly delineate these previous benchmarking efforts and their relevance to our research.
>
> **Additional references.**
>
> [1] Vasily Pestun and Yiannis Vlassopoulos. Tensor network language model. arXiv, 2017.
> [2] Daniel G. A. Smith. opt_einsum docs. https://optimized-einsum.readthedocs.io, 2018.
> [3] Devin A. Matthews. High-Performance Tensor Contraction without Transposition. In SIAM J. Sci. Comput., 2018.
> [4] Paul Springer and Paolo Bientinesi. Design of a High-Performance GEMM-like Tensor-Tensor Multiplication. In ACM Trans. Math. Softw., 2018.
> [5] Eli A. Meirom, Haggai Maron, Shie Mannor, and Gal Chechik. Optimizing Tensor Network Contraction Using Reinforcement Learning. In International Conference on Machine Learning, ICML, 2022.
> [6] John M. Martinis, Sergio Boixo, Hartmut Neven, et al. Quantum supremacy using a programmable superconducting processor. https://datadryad.org/stash/dataset/doi:10.5061/dryad.k6t1rj8, 2022.

---

### Official Review · Reviewer_zEK9 · 2024-07-23
**This paper introduced a benchmark for tensor computing libraries, testing their performance in various tensor contraction computations (einsums).**

**Rating:** 7
**Confidence:** 4
**Correctness:** Yes. The evaluation design is appropr…

**Review:**

The proposed benchmark is solid, and covers a wide enough use cases for ML practitioners. It could be a standard benchmark for future ML systems.

List of pros and cons: see sections below.

**Strengths:**

- Encompasses a wide range of tensor operations, not just deep learning.
 - A wide range of tensor libraries are tested, including commonly used NumPy, PyTorch, TensorFlow, and JAX.
 - Various different floating-point precisions are included.

**Additional Feedback:**

- L69: "the modern Einsum notation": it seems that this notation is just used within the computer science community, which may be weird to call it "modern". As far as I know physicists use the original Einstein's form (with co-variance and contra-variance indices), not the CS form.

**Clarity:**

Yes. Though some clarifications on the math notations would be better if included (in Table 1).

**Documentation:**

It'd be better to include 2-4 samples of the problem (their metadata, their Einsum equations, etc.) to make the paper more self-contained.

**Ethics:**

No.

**Limitations:**

Yes, limitations are adequately addressed.

**Opportunities For Improvement:**

- More examples would be beneficial to the readers. Including 2 - 3 sample benchmark einsum problems, with their metadata and tensor size specifications, would be appreciated. Include a figure for the contraction path.
 - Does not cover recent floating-point device features such as `bfloat16`, and `tfloat32`.
 - Table 1 would be better if common math notations are included, e.g. $\mathrm{tr}(\mathbf{A})$, $\mathbf{x}^{\rm T}\mathbf{A}\mathbf{y}$, etc.
 - Table 4 should include more problems, complexity-wise from simple to more complicated, and paired with their Einsum notations.

**Relation To Prior Work:**

Yes.

**Summary And Contributions:**

This paper curates a diverse set of use cases for the Einsum operation (tensor contraction) in machine learning, and created a benchmark to assess the performance of multiple underlying tensor libraries.

The use cases span from general linear algebra to quantum circuits, encompassing a wide range of scenarios. This benchmark provides a playground for future implementations of tensor operations, e.g. across various devices (not just GPUs).

---

> ### Author Rebuttal · Authors · 2024-08-14
>
> Thank you for providing suggestions to further improve the paper. Here are our responses to the points you have raised and how we plan to address them:
>
> **(1) Including 2 - 3 sample benchmark einsum problems, with their metadata and tensor size specifications.**
>
> Including a detailed explanation of the provided metadata for 2 - 3 actual sample benchmark problems is indeed a good idea to improve the understandability of the benchmark data set.
> However, we plan to allocate the additional content page of the paper primarily to an extra GPU experiment, expanded information on prior benchmarking efforts related to einsum, an exploration of the linkage of I/O and skewed tensor operations, and further details on why einsum is prevalent in certain domains.
> Nonetheless, we will create a separate document dedicated exclusively to describing the metadata, using examples from the benchmark dataset. Given the extensive nature of this document, it will be included in our repository to ensure it is accessible for benchmark users.
>
> **(2) Include a figure for the contraction path.**
>
> Visualizing the contraction path as a contraction tree is an effective way to enhance understanding of the computational structure of an einsum expression and how it breaks down into pairwise tensor contractions. For readers interested in this aspect, we will include explicit references to such visualizations in the background section of the paper.
>
> **(3) Does not cover recent floating-point device features such as bfloat16, and tfloat32.**
>
> We focused only on portable primitive data types to ensure that the benchmark can be utilized on a wide range of devices. However, if users require specialized data types, they can convert them from the provided ones if it is permitted by the accuracy. For example, an int16 einsum problem can be converted to a tfloat32 problem and then executed on an NVIDIA GPU.
>
> **(4) Table 1 would be better if common math notations are included.**
>
> Unfortunately, only the simple expressions can be straightforwardly expressed in common math notation. We assume most readers are familiar with mathematical expressions like matrix trace Tr(A), matrix diagonal diag(A), vector outer product uv^T, Hadamard product A∘B, and matrix chain multiplication ABCDE. Attempting to express more complex examples from the table, such as weighted model counting, graphical model query, tensor network language model query, and max-cut quantum circuit, in mathematical notation would not be straightforward and would look cumbersome, as it would also require providing tensor names. Therefore, we have included only the einsum notation to demonstrate how succinctly these problems can be formulated.
>
> **(5) Table 4 should include more problems, complexity-wise from simple to more complicated, and paired with their einsum notations.**
>
> In the data types experiment, our aim is to demonstrate that despite hardware capabilities, support for common data types is surprisingly inadequate, or implementations fail to execute at all. We believe the table is already quite detailed and sufficiently complicated, even though it features only one problem benchmarked across various data types and backends.

---

### Official Review · Reviewer_GRgD · 2024-07-26
**Nice dataset for einsums**

**Rating:** 7
**Confidence:** 4
**Correctness:** The dataset is constructed soundly an…
**Clarity:** The paper is well-written.

**Review:**

The paper is targeting an important topic, einsums, which underlie many domains, including machine learning and computational science. Improving the performance of them can have significant downstream impacts on many applications. The paper targets this by collecting a dataset of einsum expressions from a variety of domains, allowing the authors of tensor libraries (or similar) to have a uniform benchmark for comparing performance and to identify weaknesses in their implementations. The paper also evaluates the performance of common libraries for einsums on the dataset and identifies key weaknesses in them. Overall I think this is a useful dataset that significantly expands the set of einsums that prior works have considered.

That said, I think there are a few opportunities for improving the dataset and paper which might make it more impactful and better contextualize the results.

1. The dataset is fairly heavily focused on complex einsums. Indeed, it seems no problem has fewer than 26 tensors. To be more comprehensive, it may be useful to add another category of benchmarks that include simpler expressions, which may be more widely used in many areas, such as vanilla matrix-matrix multiplication. The expressions in Table 1 could easily be repurposed for this. Drawing problems from deep neural networks may be another area of opportunity.
2. The performance evaluations focus only on CPU performance, whereas many applications now use GPUs. Adding a GPU performance evaluation would be useful. (While distributed tensor algebra is very common in computational science these days, I understand skipping it as that is more complicated.)
3. The discussion in Section 4.2 on the importance of skewness and how minimizing size can improve actual performance more than minimizing flop is useful but missing some important connections. Specifically, this appears to essentially identify the importance of I/O (in the data movement sense), and that it is often the key driver of performance. This is an emerging topic in high-performance computing (although it has a long history) and there are some relevant works here. A non-exhaustive list:
    - For matrix-matrix multiplication (a core primitive for einsums), see, e.g., Kwasniewski et al., "Red-Blue Pebbling Revisited: Near Optimal Parallel Matrix-Matrix Multiplication", Supercomputing 2019; and for more general linear algebra, Kwasniewski et al., "Pebbles, Graphs, and a Pinch of Combinatorics: Towards Tight I/O Lower Bounds for Statically Analyzable Programs", SPAA 2021.
    - For tensor algebra in particular, see the works on the Cyclops Tensor Framework, e.g., Solomonik et al., "A massively parallel tensor contraction framework for coupled-cluster computations", JPDC 2014; or more recent work, e.g., Ziogas et al., "Deinsum: Practically I/O Optimal Multilinear Algebra", Supercomputing 2022.
    - In deep learning, this has also proven beneficial, see, e.g., Ivanov et al., "Data Movement Is All You Need: A Case Study On Optimizing Transformers", MLSys 2021; or many of the recent works on FlashAttention.

I mainly point this out because the paper identifies an important challenge but fails to connect it with the broader work in this area, which would be valuable for readers.

**Strengths:**

1. The paper is targeting an important topic with relevance both in machine learning and computational science.
2. The dataset constructed is (relatively, compared to other similar benchmark datasets) large and covers multiple domains.
3. The evaluation identifies weaknesses in common existing libraries, especially for einsum expressions with many tensors.
4. The dataset should be useful to those implementing tensor libraries (or similar) in providing a uniform way to compare performance and identify performance deficiencies.

**Additional Feedback:**

- The choice of the term "Language Models" for the einsum category may be a bit confusing for many people at NeurIPS, as it makes one think of transformer-style models.
- L248: While PyTorch does use ATen, in many cases for linear algebra, it will simply call standard BLAS libraries.
- In Section 4.3, the poor PyTorch performance with sparse tensors may be because it simply does not support sparse tensors in einsum.

**Documentation:**

Sufficient documentation is included.

**Ethics:**

No ethics concerns.

**Limitations:**

The paper discusses limitations adequately.

**Opportunities For Improvement:**

See above for more details, but in short:
1. It may be beneficial to include another benchmark category focusing on simpler, more "standard" einsums.
2. Can a GPU performance evaluation be added?
3. A discussion of I/O complexity may be useful to add to Section 4.2.

**Relation To Prior Work:**

The paper does not really discuss prior work in benchmarking einsums (if any) and their performance. While there may not be explicit datasets/benchmarks (that I know of), prior works have constructed various evaluations although they are not fully comparable. A discussion of these could be helpful; the above references on multilinear algebra are a good starting point.

**Summary And Contributions:**

The paper constructs a dataset of einsum expressions for use in evaluating the performance of different einsum implementations (e.g., tensor libraries). The expressions are drawn from a variety of areas, including machine learning and computational physics, and differ significantly from the typical einsums seen in libraries (e.g., simple matrix-matrix multiplies). The einsum expressions are analyzed to collect various metadata. Finally, the performance of some typical einsum evaluation setups is evaluated, and bottlenecks related to, e.g., large numbers of tensors are identified, as are limitations related to einsum constructs that are not well-supported.

---

> ### Author Rebuttal · Authors · 2024-08-14
>
> Thank you for your insightful feedback. Here is how we plan to address the points you have raised:
>
> **(1) Simpler einsum expressions.**
>
> As noted in the limitations section of our paper: "We only included problems in the dataset that start small and become large when evaluating the einsum expression. Otherwise, the benchmark dataset would quickly exceed the limits of common storage devices and require long read times for the data, which would reduce its practicability." Simple einsum expressions that are computationally intensive typically require substantial data for the tensors right from the start. For practical reasons, we decided to omit such expressions from the benchmark to ensure the dataset remains manageable and accessible.
>
> **(2) GPU experiment. (Note: This suggestion is similar to one raised by Reviewer 3ZT8)**
>
> Yes, insights into the performance of einsum expressions on GPUs would complete the experiments section. We will add a fifth experiment to the paper that specifically targets performance pitfalls on GPUs. From our work with einsum, we know that although a single matrix-matrix multiplication often performs better on a dedicated GPU than on a CPU, executing the entire einsum expression on a GPU can be significantly slower. We will investigate this peculiarity further in an experiment.
>
> **(3) I/O complexity for skewed tensor operations.**
>
> Your comments on how I/O and asymmetric tensor operations relate, along with the referenced works, could be immensely helpful for developers trying to devise strategies to speed up skewed operations in einsum. We will incorporate these insights into Section 4.2.
>
> **(4) Prior work in benchmarking einsum. (Note: This suggestion is similar to one raised by Rewiever 3ZT8)**
>
> Existing benchmarks have primarily focused on structurally simple random einsum expressions (opt_einsum [1]) or those involving only a few random tensors (TBLIS [2], GETT [3]). Additionally, there is notable benchmarking related to contraction path computations, particularly within the realm of quantum computing, which primarily assesses theoretical metrics rather than execution. We have integrated instances from these quantum computing benchmarks or used their generation methods [4, 5]. In the final version of our paper, we will clearly delineate these previous benchmarking efforts and their relevance to our research.
>
> **(5) PyTorch performance with sparse tensors may be low because it simply does not support sparse tensors in einsum.**
>
> This is correct and should be mentioned explicitly so that it is clear to every reader.
>
> **Additional references.**
>
> [1] Daniel G. A. Smith. opt_einsum docs. https://optimized-einsum.readthedocs.io, 2018.
> [2] Devin A. Matthews. High-Performance Tensor Contraction without Transposition. In SIAM J. Sci. Comput., 2018.
> [3] Paul Springer and Paolo Bientinesi. Design of a High-Performance GEMM-like Tensor-Tensor Multiplication. In ACM Trans. Math. Softw., 2018.
> [4] Eli A. Meirom, Haggai Maron, Shie Mannor, and Gal Chechik. Optimizing Tensor Network Contraction Using Reinforcement Learning. In International Conference on Machine Learning, ICML, 2022.
> [5] John M. Martinis, Sergio Boixo, Hartmut Neven, et al. Quantum supremacy using a programmable superconducting processor. https://datadryad.org/stash/dataset/doi:10.5061/dryad.k6t1rj8, 2022.

---

> > ### Comment · Reviewer_GRgD · 2024-08-16
> >
> > > For practical reasons, we decided to omit such expressions from the benchmark to ensure the dataset remains manageable and accessible.
> >
> > This makes sense, although I would note that for these sort of benchmarks, you could probably just deterministically generate the input at runtime rather than distributing the data.
> >
> > > We will add a fifth experiment to the paper that specifically targets performance pitfalls on GPUs.
> >
> > Thanks, this should be a useful component.
> >
> > > From our work with einsum, we know that although a single matrix-matrix multiplication often performs better on a dedicated GPU than on a CPU, executing the entire einsum expression on a GPU can be significantly slower. We will investigate this peculiarity further in an experiment.
> >
> > This is likely due to excess data movement caused by the einsums being implemented as a relatively naive sequence of matrix-matrix multiplies. I think it is good to point this out as it should help drive further development.
> >
> > Overall, I remain positive on the paper and continue to recommend acceptance.

---

### Official Review · Reviewer_Su7F · 2024-08-02

**Rating:** 8
**Confidence:** 4
**Correctness:** I believe the paper is sound.
**Clarity:** Well-written.

**Review:**

Thank you for submitting your work to the dataset and benchmarking track. I am happy to see a dataset of einsum expressions used in various domains from quantum computing to sat solving etc. This further covers many interesting cases of einsum variations. Cases where the same dimension is used for contractions over multiple tensors and also expressions that are multi-dimensional. Usually, program optimization work on tensor computations is evaluated in 2D or 3D, not the higher dimensions presented in this dataset. As a result, I expect newer techniques to be invented because of this dataset. They can be theoretical or empirical program optimization work including the potential for programming languages work. Additionally, I noticed that authors are even including sparse tensor algebra, which is a growing area of interest in many domains. The paper further points out the limitations of existing programming frameworks which is a plus. Overall, I really enjoyed reading this paper and this would be valuable contribution to the community.

**Strengths:**

* Well-written
* Diverse dataset covering various domains and interesting cases of einsum notations
* Good exploration of limitations of exisiting frameworks.

**Additional Feedback:**

None.

**Documentation:**

Yes

**Limitations:**

Mentioned in the paper.

**Opportunities For Improvement:**

The paper is already well-written. One place you can improve would be to give more detail of some of the applications and why particular einsums exist in those domains.

**Relation To Prior Work:**

Yes

**Summary And Contributions:**

The paper introduces a benchmark suite of tensor problems that can be expressed as einsums. In my opinion, this is a unique benchmark suite that can help program optimization domain. There can be theoretical work, optimization work, and programming language work that can use this benchmark suite. Hence, I find it intriguing and helpful to the community.

---

> ### Author Rebuttal · Authors · 2024-08-14
>
> Thank you for your comments and asking for more details of why einsum exist in certain domains.
>
> Einsum offers a straightforward interface for describing tensor computations, by using just a format string that outlines the operation and a list of tensors. Before it was recognized that einsum alone was sufficient for contracting tensor networks, it was common to utilize specialized tensor network libraries. These libraries required describing the nodes and their connections in a graph-like manner, as seen in Google's tensor network library [1, 2]. The introduction of einsum has simplified the execution of tensor networks and tensor network algorithms. Moreover, unlike traditional tensor networks, which are structurally simple and lack features like hyperedges or Hadamard products, einsum can handle these and other complexities. For instance, tensor networks often require the use of copy-tensors [3] to manage hyperedges, increasing memory usage and reducing performance, whereas einsum eliminates the need for such workarounds. Its versatility makes einsum particularly advantageous in fields that use tensor networks, such as AI/ML and quantum computing, because it is more expressive and allows expressing computations more succinctly.
>
> Apart from handling complex tensor network expressions, einsum has also become well-established in the Deep Learning community for expressing relatively small tensor operations. It aids in formulating tensor operations more clearly and succinctly than has been possible with previous library routines [4].
> We will include these explanations in the paper to detail why einsum has proliferated across domains that utilize tensor networks and deep learning.
>
> **Additional references.**
>
> [1] Chase Roberts, et al. TensorNetwork: A Library for Physics and Machine Learning. arXiv, 2019.
> [2] TensorNetwork Developers. TensorNetwork Software. https://github.com/google/TensorNetwork, 2019.
> [3] Jacob D. Biamonte, Jason Morton, and Jacob W. Turner. Tensor Network Contractions for #SAT. Journal of Statistical Physics, 2015.
> [4] Tim Rocktäschel. Einsum Is All You Need - Einstein Summation in Deep Learning. https://rockt.github.io/2018/04/30/einsum, 2018.

---

### Decision · Program_Chairs · 2024-09-26

**Decision:**

Accept (Poster)

**Comment:**

The paper proposes a benchmark framework for tensor execution engines based on Einsum, a widely-used interface for describing tensor expressions. The reviewers unanimously acknowledged the framework's uniqueness, extensiveness, and practical significance, indicating that the paper is well-deserving of acceptance.

The reviewers also offered constructive suggestions, including experiments on GPUs, discussions on I/O and skewed tensor operations, and examples for improving clarity. The authors are expected to address these points in the camera-ready version.